# Personality as a Resource for Labor Market Participation among Individuals with Chronic Health Conditions

**DOI:** 10.3390/ijerph17176240

**Published:** 2020-08-27

**Authors:** Sandra Brouwer, Sander K. R. van Zon, Ute Bültmann, Harriëtte Riese, Bertus F. Jeronimus

**Affiliations:** 1Community and Occupational Medicine, Department of Health Sciences, University Medical Center Groningen, University of Groningen, Hanzeplein 1, P.O. Box 30.001, 9700 RB Groningen, The Netherlands; s.k.r.van.zon@umcg.nl (S.K.R.v.Z.); u.bultmann@umcg.nl (U.B.); 2Interdisciplinary Center Psychopathology and Emotion regulation (ICPE), Department of Psychiatry, University Medical Center Groningen, University of Groningen, Hanzeplein 1, P.O. Box 30.001, 9700 RB Groningen, Groningen, The Netherlands; h.riese@umcg.nl (H.R.); b.f.jeronimus@rug.nl (B.F.J.); 3Department of Developmental Psychology, University of Groningen, Grote Kruisstraat 2/1, 9712 TS Groningen, The Netherlands

**Keywords:** personality, employment, chronic diseases, NEO-PI-R, work

## Abstract

*Background:* The link between personality traits and employment status in individuals with chronic health conditions (CHCs) is largely unexplored. In this study, we examined this association among 21,173 individuals with CHCs and whether this association differs between individuals suffering from a heart disease, depression, anxiety, cancer, chronic obstructive pulmonary disease, musculoskeletal disease (MSD) and type 2 diabetes mellitus (T2DM). *Methods:* This study was conducted using baseline data from the Lifelines Cohort Study. Employment status and the presence of CHCs were determined by questionnaire data. The Revised Neuroticism-Extroversion-Openness Personality Inventory (NEO-PI-R) was used to measure eight personality facet traits. We conducted disease-generic and disease-specific logistic regression analyses. *Results:* Workers with higher scores on self-consciousness (OR: 1.02; 95% CI: 1.01–1.02), impulsivity (1.03; 1.02–1.04), excitement seeking (1.02; 1.01–1.02), competence (1.08; 1.07–1.10) and self-discipline (1.04; 1.03–1.05) were more often employed. Adults with higher scores on anger-hostility (0.97; 0.97–0.98), vulnerability (0.98; 0.97–0.99), and deliberation (0.96; 0.95–0.97) were least often employed. Personality facets were associated strongest with employment status among individuals suffering from MSD and weakest in individuals with T2DM. *Conclusions:* Personality might be a key resource to continue working despite having a CHC. This may be relevant for the development of targeted personality-focused interventions.

## 1. Introduction

The growing proportion of older workers in the labor market stresses the need to promote a healthy working life throughout a person’s life cycle [1]. Up to 30% of the European working population reports a chronic health condition (CHC) [2,3]. Together with self-perceived poor health, CHCs are important risk factors for disability pension, unemployment, and, to a lesser extent, early retirement [4]. Many workers with a CHC leave the labor market prior to retirement age [5,6], but some successfully continue their work roles. Successful job role performances in workers with a CHC have been related to illness acceptance, coping behavior, and finding meaning [7], as well as job characteristics including high autonomy, high flexibility, low work pressure, and high support from colleagues [8,9]. The relation between these factors and work participation was found in workers with different types of CHCs, which suggests that these factors are disease generic [10].

A more specific individual resource that is understudied in the context of labor market participation is personality, which captures characteristic differences in what workers feel, think, want and do [11]. Personality traits are confined suits of cognitive-behavioral traits that vary consistently amongst individuals, but are maintained through time and across context. Personality differences have previously been linked to the development of CHC, which in turn can influence personality development, via the experienced limits and constraints in functioning and activities [12,13,14]. Moreover, in the context of CHCs, personality traits can be both risk and resilience factors [15]. Individuals who develop CHCs and continue their work roles may have different personality traits than their peers who leave the labor market, receive disability benefits, or become unemployed. The identification of personality traits as resilience factors linked to employment status of individuals with CHCs may enable the development of targeted intervention strategies. Personality-tailored interventions proved to be highly effective strategies to influence problematic coping behaviors, including interpersonal dependence, avoidance, aggression, risky health behaviors and substance misuse [16,17,18].

The paradigmatic and currently most used personality taxonomy captures individual differences in five higher-order personality factors (i.e., extraversion, agreeableness, conscientiousness, neuroticism, and openness to experience), where each subsume a set of more specific lower-order facet-level personality traits [11,19,20]. Lower levels of neuroticism and higher levels of extraversion and conscientiousness have been associated with more successful work role performances over the lifespan [21,22,23,24]. Higher neuroticism has been associated with more experienced stress and higher stress reactivity, less acceptance and problem solving, more avoidance, more mental problems [23,25], lower vigor and determination [24], and poorer job performance [26], as the capacity to allocate resources to accomplish tasks tends to be impeded in those individuals. More extraverted individuals are typically more status and power focused, show more work engagement [24] and may experience a stronger urge to maintain their social contacts at work and to derive novel experiences, which in turn may motivate them to continue to work and put up with demands and efforts. Highly conscientious individuals typically show more thoroughness, self-discipline, work-role investment, adherence to rules [24,27], and more adaptive coping strategies [23], which results in superior planning and job performance [26,28].

There is a body of literature on the link between personality and employment in population samples [29,30], but the association between personality facet traits and employment in people with a CHC remains virtually uncharted territory (see for exceptions, e.g., Krause et al. [31]). Personality facets are most relevant and informative from an intervention perspective, as these etiologically distinct entities [32] capture more confined and realistic targets, and are more likely to be predictive *and* explanatory in the work context [28,33]. The link between personality facets and employment status among workers with CHCs is therefore an important topic of inquiry from both a practical and theoretical perspective.

The present study expands the literature with tests of the available facet traits within the neuroticism, extraversion, and conscientiousness domains on employment status among workers with a CHC. Our a priori hypotheses in Table 1 were based on the known associations between higher-order personality domains and work characteristics [21] and the definitions and specific content of the facet traits [20]. Specifically, this study aims to (a) examine the association between personality facets and employment status among individuals with CHCs and (b) investigate whether this association differs across the most common CHCs in Western societies, which are heart disease, depression, anxiety, cancer, chronic obstructive pulmonary disease (COPD), musculoskeletal disease (MSD) and type 2 diabetes mellitus (T2DM) [34,35].

## 2. Materials and Methods

### 2.1. Design and Sample

This study was conducted using data from the Lifelines Cohort Study [36,37]. Lifelines is a multi-disciplinary prospective population-based cohort study examining the health and health-related behaviors of the population living in the three northern provinces of The Netherlands. Lifelines assesses biomedical, sociodemographic, behavioral, physical and psychological factors which contribute to the health and disease of the general population. Recruitment and data collection have been described extensively elsewhere [37]. In short, participants were recruited through invitations by their general practitioner or family members. In addition, there was an option to self-register. Lifelines was conducted according to the guidelines in the Declaration of Helsinki and all procedures involving human subjects were approved by the Medical Ethics Committee of the University Medical Center Groningen. Written informed consent was obtained from all participants. The current study uses data from adult participants between age 18 and 65 years (of working age) who visited the research centers between November 2006 and March 2013 for the baseline measurements.

### 2.2. Measures

#### 2.2.1. Employment Status

Employment status was assessed with the following question: “Which situation applies to you?” (answer categories: work ≥ 32 h per week; 20 ≤ work< 32 h per week; 12 ≤ work <20 h per week; work < 12 h per week; early retirement; unemployed/looking for work; disabled; welfare; homemaker; student). Those working ≥ 12 h per week were categorized as employed, those working < 12 h per week, unemployed, disabled, on welfare, homemakers and with early retirement were categorized as non-employed. Students were excluded from the analyses because they are not available for the labor market.

#### 2.2.2. Chronic Health Conditions

The presence of CHCs was determined by self-reported questionnaire data. The included CHCs, i.e., cancer, T2DM, COPD, heart disease, MSD, depression and anxiety, were based on the ranking of leading causes of disability adjusted life years (DALYs) in Western Europe and the Netherlands [34,35]. Cancer was assessed with the following question: “Do, or did, you have cancer?” (answer categories: yes/no). T2DM was assessed with the following question: “Do you have diabetes mellitus?” (yes/no), and a follow-up question determined the type of diabetes. COPD was assessed with the following question: “Do you have COPD, emphysema or chronic bronchitis?” (yes/no). Heart disease was determined by the following questions: “Did you ever had a myocardial infarction?”; “Were you ever diagnosed with an aortic aneurysm?”; “Do you have heart failure?”; “Have you ever had a balloon angioplasty and/or bypass surgery?” (yes/no). If one of these four questions was answered positively, participants were categorized as having heart disease. MSD was determined by the following questions: “Do you have rheumatoid arthritis” (answer categories: yes; no) and “Could you indicate which of the following disorders you have (had)?” (answer category included osteoarthritis). If one of these questions was answered positively, participants were categorized as having a musculoskeletal disorder. Depression and anxiety disorders were assessed with the following question: “Could you indicate which of the following disorders you have (had)?” (answer category included depression and (other) anxiety disorder).

#### 2.2.3. Personality

The Big Five personality traits are overarching domains that contain and subsume most relevant variation in adult personality traits and seem to represent the basic structure behind our personality traits as they emerge in our natural language [11]. The Big Five trait scores were captured with the revised Neuroticism-Extroversion-Openness (NEO) Personality Inventory (NEO-PI-R) which comprises 30 lower-order ‘facet’ traits, which reflect more specific aspects within each of the five broader personality domains [19]. In the present study, we examined eight personality facets across three higher-order domains of Neuroticism (i.e., anger-hostility, self-consciousness, impulsivity, and vulnerability), Extraversion (excitement seeking), and Conscientiousness (viz., competence, self-discipline, and deliberation). These eight personality facets were selected based on their influence on psychopathology (according to the NEO manual) and were defined and illustrated with sample items in Table 1.

All 64 items were answered on a 5-point Likert-type scale that ranged from strongly disagree to strongly agree. Each facet scale score reflects the sum of scores on the 8 corresponding items, thus ranges from 8 to 40. Sum scores were calculated when ≥5 valid item scores were available, and adjusted for the number of missing items. The authorized Dutch translation of this scale has good reliability and validity [38]. For each subscale of the NEO, we also obtained reliability information by calculating Cronbach’s alpha. Cronbach’s alpha’s for the facets self-consciousness, self-discipline and vulnerability ranged from ≥0.8 to 0.9, Cronbach’s alpha’s for the facets competence, anger-hostility and deliberation ranged from ≥0.7 to 0.8 and Cronbach’s alpha’s for the facets impulsivity and excitement seeking ranged from ≥0.6 to 0.7. Table 1 shows the correlation between the personality facets.

#### 2.2.4. Covariates

Age, gender, marital status, and educational level were taken into account as basic sociodemographic factors. Age was calculated based on the date of the first clinical visit. Marital status was categorized as living together or not living together. Educational level was operationalized as the highest educational level achieved, categorized into low (no education, primary education, basic vocational education, secondary education), medium (senior secondary vocational education, general senior secondary education), and high (higher professional education, higher academic education). Multi-morbidity and self-rated health were taken into account as crude indicators for disease severity. Multi-morbidity was defined as having ≥2 chronic health conditions. Self-rated health was measured on a 5-point Likert scale and was categorized into good (good, very good, excellent) and poor (fair, poor) self-rated health.

### 2.3. Statistical Analysis

First, we described the baseline characteristics of employed and non-employed participants with a CHC using descriptive statistics, including percentages and group means with their standard deviation (SD). Differences in baseline characteristics of employed and non-employed participants in terms of standardized mean difference *d* [39], heuristically interpreted as small when ranging from 0.20 to 0.40, medium from 0.41 to 0.79, and large when >0.80. Because the compared groups have different sample sizes, the calculated pooled SD was adjusted with weights for the sample sizes and confidence intervals were calculated according to Hedges and Olkin [40]. Second, we examined the association between personality facets and employment status using a disease-generic logistic regression analysis with non-employed as the reference category. Odds are defined as the ratio of the probability of success and the probability of failure and range between 0 and infinity. We adjusted for the sociodemographic factors age, gender, marital status and educational level. To examine whether the effect of personality facets traits on employment status differs between CHCs we repeated the analyses for each specific CHC. In sensitivity analyses, we examined whether multi-morbidity and self-rated health had any influence on the associations between personality facet traits and employment status. First, our models were adjusted for the presence of multi-morbidity and, subsequently, also for self-rated health. All analyses were performed using SPSS (version 22; IBM Corp., Armonk, NY, USA). Tests were considered significant at *p* < 0.05 but because of the size of our sample and exploratory nature of our study, we focused on estimates significant at *p* < 0.01 or smaller.

## 3. Results

### 3.1. Baseline Characteristics

A total of 21,173 participants of working age with a CHC were included in this study (See flowchart in Appendix A), of which 14,915 (70.4%) were employed (Table 2). Non-employed participants with a CHC were on average older, more often female, lower educated, and reported worse health compared to employed participants with a CHC. The prevalence of T2DM, depression, anxiety, and MSD was higher among the non-employed compared to the employed participants, while the prevalence of COPD was lower.

Among people with a CHC, employed and non-employed participants differed significantly on all the NEO personality facet scales (see Table 2). Employed participants reported higher scores than non-employed participants on impulsivity (*d* = 0.05, with a 95% confidence interval [0.02–0.08])), excitement seeking (*d* = 0.33 [0.30–0.36], from the extraversion domain), and on competence (*d* = 0.41 [0.38–0.44), self-discipline (*d* = 0.28 [0.25–0.31]) and deliberation (*d* = 0.05 [0.02–0.08], from the conscientiousness domain). Employed participants reported lower anger-hostility (*d* = 0.18 [0.15–0.21]), self-consciousness (*d* = 0.22 [0.19–0.25]) and vulnerability (*d* = 0.37 [0.34–0.40]—all three from neuroticism domain). These differences between employed and non-employed participants were small to moderate in size.

### 3.2. Personality Facets and Employment Status: Disease-Generic Analysis

In the *neuroticism* domain, higher scores on self-consciousness (OR: 1.02; 95% CI: 1.01, 1.02) and impulsivity (1.03; 1.02, 1.04) were associated with higher odds for being employed, whereas higher scores on anger-hostility (0.97; 0.97, 0.98) and vulnerability (0.98; 0.97, 0.99) were associated with lower odds for being employed (Table 3). In the *extraversion* domain, a higher score on excitement seeking (1.02; 1.01, 1.02) was associated with a higher odds for being employed. In the *conscientiousness* domain, higher scores on competence (1.08; 1.07, 1.10) and self-discipline (1.04; 1.03, 1.05) were associated with higher odds for being employed, whereas a higher score on deliberation (0.96; 0.95, 0.97) was associated with a lower odds for being employed.

### 3.3. Personality Facets and Employment Status: Disease-Specific Analyses

Associations between personality facets and employment status differed across the health conditions (Table 4). For *neuroticism,* a higher score on self-consciousness was associated with higher odds for being employed in individuals reporting COPD. A higher score on impulsivity was associated with higher odds for being employed in individuals reporting depression, heart disease, COPD, and MSD. A higher score on anger-hostility was associated with lower odds for being employed in individuals reporting cancer, depression, an anxiety disorder, COPD and MSD. Further, a higher score on vulnerability was not associated with a lower odds for being employed. For *extraversion*, a higher score on the excitement seeking facet was associated with a higher odds for being employed when being depressed. For *conscientiousness*, higher scores on competence and self-discipline were associated with higher odds for being employed across all CHCs, except for T2DM. Finally, a higher score on deliberation was associated with a lower odds for being employed across two CHCs, namely, depression, and MSD. Table 5 gives an overview of the associations in relation to the a priori hypotheses.

### 3.4. Sensitivity Analyses

In the disease-generic analysis (Appendix A), the associations between personality facets and employment status remained statistically significant after adding multi-morbidity (model 1) and additionally self-rated health (model 2) to the original model, except for vulnerability (at *p*-level < 0.01 in model 2). In the disease-specific analyses (Appendix A)*,* three out of 36 associations were no longer statistically significant (*p* < 0.05) after adding multi-morbidity to the model. The association between vulnerability, from the *neuroticism* domain, and employment status was no longer statistically significant among individuals with an anxiety disorder or MSD. The association between self-consciousness, also in the domain of *neuroticism*, and employment status also disappeared among individuals with a MSD. For eleven associations, the significance level was no longer <0.01, but still under *p* < 0.05. After adding self-rated health (Appendix A), another four associations disappeared. Anger-hostility, also in the domain of *neuroticism*, was no longer associated with employment status among individuals with depression and COPD. Deliberation, in the domain of *conscientiousness*, was no longer associated with employment status among individuals with an anxiety disorder or COPD. In addition, for another three associations, the significance level was no longer <0.01, but still under *p* < 0.05.

## 4. Discussion

This study showed that all facet traits within the personality domains neuroticism, extraversion, and conscientiousness were associated with the employment status of individuals with CHCs. Individuals with higher scores on competence, self-consciousness, impulsivity, excitement seeking and self-discipline were more often employed. Individuals with higher scores on anger-hostility, vulnerability and deliberation, in contrast, were more often inactive in the labor market, e.g., working less than 12 h, homemakers, unemployed or receiving disability benefits.

Some disease-generic factors were observed, as across CHCs, working 12 h or more was associated with more self-reported competence, impulsivity and self-discipline. Not all associations between facets and employment status remained significant in the context of all CHCs, thus some processes seem disease specific; deliberation was associated with employment states in individuals with depression and MSD only. Moreover, for some CHCs, most of the personality facets were linked to employment status, i.e., MSD (5 out of 8), depression (6/8), and COPD (5/8), while in people with heart diseases, only 3 out of 8 facets were associated and for T2DM, none of the facets. Note that our interpretation of these associations was somewhat conservative (*p* < 0.01 or smaller). For example, some trends could be observed for vulnerability. Future work may replicate and substantiate these observations.

Most of the associations between the personality facets and employment status were in keeping with our a priori expectations as formulated in Table 1, with notable exceptions for self-consciousness and impulsivity (both facets traits of neuroticism) and deliberation (facet trait of conscientiousness). Higher levels of self-consciousness (which taps into proneness to shame and feeling uncomfortable in company) and impulsivity (or inability to control cravings and urges) were associated with higher odds for being employed in the context of CHCs. A higher level of deliberation indicates being more cautious, deliberate, and considerate, which was associated with lower odds of being employed. To the best of our knowledge, there is no research that tapped into possible underlying mechanisms that can help to explain these results. However, these results underscore the importance of focusing on the facet level to understand the consequences of CHCs, as some observed associations for the facets oppose those of their overarching higher-order domain, as has previously been observed for normative developmental trajectories [41].

In line with our expectations in Table 1, in a study among healthy adults, deliberation has been found to be associated with higher job stability, whereas impulsivity was associated with higher job instability [42]. Apparently, these personality facets show different, or even opposite, associations in the context of CHCs, although replication remains warranted, also to unravel underlying processes. Nonetheless, it is tempting to speculate that in most people, CHCs increase self-focus, and the unique problem-solving coping strategies associated with deliberation [23] may result in giving up one’s work role, as a self-protective strategy to prioritize their private life when balancing the demands of the illness (e.g., time, energy, money and social support [43,44]). This would align with the notion that more conscientious individuals are more likely to visit doctors and to adhere to medical advice [45]. Impulsivity captures aspects of boldness and risk taking (neurotic immoderation), and is the opposite of the careful thinking and planning tapped into by the deliberation facet [46]. More impulsive people, in contrast, may not be able to resist going to work (via the needs work satisfies and inclination to wishful thinking [23]), even though they know that they should refrain from doing so [19], which may further deteriorate their health status [22]. In analogy, high self-consciousness may motivate people to persist in their work role in the face of CHCs to deviate their attention from inner states and to fulfil perceived expectancies and perceptions by others [47], even at high personal costs. Importantly, facet trait scores are also likely to link individuals to job types and associated differences in, among others, autonomy, flexibility, physical workload, status, team work, and job satisfaction [21,48,49], which may play a role in the observed outcomes. For example, if more deliberate individuals had more cognitively taxing, competitive and achievement-oriented jobs, this may have made it more difficult for them to persist when suffering from CHCs. Future longitudinal studies could explore these possibilities in more detail and control for alternative explanations, from differences in social networks to interactions with contextual features [48].

One implication of these speculations would be that more people could persist in their work roles despite their CHCs if they had more freedom to manage the physical, social, or organizational aspects of their job according to their individual limitations in order to balance the demands of the illness against those of work and private life [43,44]. Several studies have shown that various forms of employer support such as job modification, offering flexible work schedules, provision of special equipment and offering training seem effective in retaining workers with CHCs in work [50,51]. Future research should not only focus on work environment, but also on potential intervention strategies that influence health via personality. Interventions that target personality facets such as anger-hostility and impulsivity are known to improve self-management, which is likely to improve health and happiness in the context of CHCs [16,18]. Interventions could also aim to foster competence and self-discipline to influence core behaviors that underlie the manifestation of these traits with the goal of engendering new, healthier patterns of behavior that, over time, become automatized and translate into personality changes [52]. Such interventions could also be CHC specific, as a better regulation of diabetes or other somatic disorders could have the additional benefit of changing one’s personality [16,53]. Finally, supervisors can support employees with low conscientiousness by helping them plan and structure and create reminders, which may prevent the emergence of CHCs and drop out once they surface [45].

### 4.1. Strengths and Limitations

The results of this study should be interpreted in light of the following strengths and limitations. A strength is that our large population sample enabled precise estimates of the association between personality facet traits and employment status across CHCs. This demonstrated that associations may be different between different CHCs. The Lifelines Cohort Study population has been found to be representative of the general Dutch population [54], which allows generalizability of the findings to the Netherlands. Second, the NEO facet scales are part of a reliable and validated instrument [38], and their association with employment status when faced with a CHC seems to have escaped attention and this is the first study to address this topic.

A major limitation is that cross-sectional associations do not inform about causation, and do not support clear suggestions for intervention strategies. Nevertheless, it is generally believed that personality typically changes slowly (over months and years), which would limit the possibility of reverse causation [55], although some studies showed that declines in health status may indeed lead to changes in dispositional characteristics such as the Big Five personality traits [12,53]. Moreover, specific personality traits also increase the risk of developing CHCs [14,56]; differences in neuroticism and conscientiousness, for example, predict anxiety disorders and major depression [25,57], as well as Alzheimer’s disease [58,59]. Second, all diagnoses of the CHCs were based on self-reports, which may include some bias that may have influenced our findings. However, previous research has shown that self-reports on CHCs are fairly accurate as compared to information from the general practitioner [60].

Finally, it is important to realize that the associations between personality facets and employment status in the face of CHCs are likely to differ across occupational groups (e.g., whether one’s job is task or other oriented) [21], which makes our estimates somewhat crude. The process of accommodation to one’s CHC over time has specific phases in which different traits may play a key role [51]. Future studies should therefore account for combinations of specific CHCs, personality facets, and contexts, ideally at the individual level, to better understand the underlying processes [61] and to focus on disease-specific challenges and limitations.

### 4.2. Implications

This study provides new and unique insights into associations between personality facet traits and employment status of individuals with a CHC. Understanding how the employment rates of individuals with CHCs can increase is essential in the context of the societal debate on increasing retirement ages and prolonging work life of all people, including those employed with CHCs. Being chronically ill is not a static matter but a continuous process of balancing the demands of the illness and the demands of everyday (work) life [43]. Adaptations to health problems form a process between and within people and circumstances and depend on individual’s characteristics, the nature of his or her work, and relationships with others [9]. The findings from this study can be relevant and informative for the development of targeted interventions for individuals with a CHC to enhance labor market participation of these vulnerable workers. Yet, longitudinal studies are necessary to inform these future steps in more detail. Since personality may change over time and the age at which individuals develop a CHC may affect how individuals perceive and cope with the CHC, we recommend that future studies investigate possible age effects in the association between personality facet traits and labor market participation among individuals with a CHC. In addition, examining the mediating role of healthy behaviors and job characteristics on this association might be relevant, as both might be influenced by personality [14,49] and are known to increase the risk of early exit from paid employment of individuals with a CHC [62,63]. Finally, future studies should include more personality facet traits—for example, facets in the openness to experience domain (e.g., creativity), which have been implicated in health and longevity [64] and openness plays a role in the burden of disease and job functioning via communication and disclosure with one’s supervisor and physician [65] and enhanced creativity and innovation [48].

A theoretical implication follows from our models using eight different facet traits. The associations between the diverse facet traits from the neuroticism or conscientiousness domains and work outcomes in the context of CHCs should have been approximately equal to purport causality at the trait level [61]. Our models in which highly divergent associations were observed between facet traits from the same domain indicate that facets (or items), and not the broad traits, are associated with the ability to persist in the work role. This conclusion has implications for inferences from our work and the previous studies at the domain level—as outlined in our introduction section—as some facets are clearly more central to the functioning of people with CHCs than others. These facets deserve more attention when one aims to develop prevention and intervention strategies in the context of CHCs.

## 5. Conclusions

Our findings suggest that one’s personality might be a key resource to continue working despite having a CHC. In particular, competence, impulsivity and self-discipline were found to be associated with employment status across different types of diseases. Vulnerability and deliberation were only relevant in a specific disease. This may be relevant for the development of targeted personality-focused interventions. Future studies of overlap and differences in the associations between personality facet traits and CHCs may hint at their developmental trajectories. Therefore, longitudinal studies are needed to further examine the specific trajectories before firm conclusions can be drawn.

## Figures and Tables

**Table 1 ijerph-17-06240-t001:** A priori hypotheses about the association between personality and employment.

Personality Domain and Facet Traits	Definition of High Score	Expected Association on Employment Status
*Neuroticism*	Prone to negative emotions and pessimism	
Anger-hostility	Angry, irritable, frustrated, and bitter	−
Self-consciousness	Prone to shame, embarrassment, and feeling uncomfortable and inferior in company	−
Impulsivity	Inability to control cravings and urges	−
Vulnerability	Dependent, hopeless and panicky when stressed	−
*Extraversion*	Sociability, energy and optimism	
Excitement seeking	Search for new experiences and feelings	+
*Conscientiousness*	Organized, planful, and self-disciplined	
Competence	Capable, sensible, prudent, and effective	+
Self-discipline	Task focus, perseverance, and self-control	+
Deliberation	Cautious, deliberate, and considerate	+

Note: Descriptions derived from Costa and McCrae (2006). Low scores can be inferred as the reverse/opposite of a high score. Data on the facets included in this table were the only available personality traits in the Lifelines Cohort Study; see method section for further details. The ‘–’ indicates that the higher the scores on these scales, the more likely these people are to become unemployed when faced with a chronic health condition (CHC), whereas ‘+’ means that higher scores are expected to be associated with a higher odds of being employed while having a CHC.

**Table 2 ijerph-17-06240-t002:** Baseline characteristics.

Characteristics	Total Sample *N* = 21,173	Employed *N* = 14,915	Non-Employed *N* = 6258	*p*-Value
Age, mean (SD)	46.4 (10.0)	45.0 (9.2)	49.8 (11.2)	<0.001
Gender, female, %	65.5	39.4	77.0	<0.001
Living together, %	79.9	80.8	77.5	<0.001
Educational level, %				
High	24.8	29.1	14.6	
Medium	38.8	41.6	32.2	<0.001
Low	36.4	29.4	53.3	
Personality facets, mean (SD)				
Anger-hostility	19.9 (4.6)	19.6 (4.5)	20.4 (4.7)	<0.001
Self-consciousness	20.3 (5.0)	20.0 (4.9)	21.1 (5.2)	<0.001
Impulsivity	22.8 (4.0)	22.8 (4.0)	22.6 (4.0)	0.002
Vulnerability	19.5 (4.7)	19.0 (4.5)	20.7 (5.0)	<0.001
Excitement seeking	21.3 (4.6)	21.8 (4.5)	20.3 (4.5)	<0.001
Competence	28.9 (3.7)	29.3 (3.6)	27.8 (3.8)	<0.001
Self-discipline	28.5 (4.7)	28.9 (4.5)	27.6 (5.0)	<0.001
Deliberation	27.9 (4.4)	27.9 (4.3)	27.7 (4.5)	<0.001
Multi-morbidity, %				
No	79.8	83.3	71.5	<0.001
Yes	20.2	16.7	28.5	
Chronic diseases, %				
Cancer	13.8	13.8	13.9	0.914
T2DM	4.9	4.0	7.0	<0.001
Depression	39.2	37.7	42.8	<0.001
Anxiety disorder	12.6	12.0	13.8	0.001
Heart disease	5.9	5.4	7.0	<0.001
COPD	19.1	20.0	17.0	<0.001
Musculoskeletal disorder	28.5	26.1	34.1	<0.001

Abbreviations: SD: standard deviation; T2DM: type 2 diabetes mellitus; COPD: chronic obstructive pulmonary disease.

**Table 3 ijerph-17-06240-t003:** Associations of personality and employment status ^1^.

Characteristictics	Odds Ratio (95% Confidence Interval)
Age	0.95 (0.95, 0.95) ***
Gender	
Female	Ref
Male	2.25 (2.08, 2.42) ***
Marital status	
Living together	Ref
Not living together	0.69 (0.63, 0.75) ***
Educational level	
High	Ref
Medium	0.68 (0.62, 0.75) ***
Low	0.36 (0.33, 0.39) ***
Personality facets	
Anger-hostility	0.97 (0.97, 0.98) ***
Self-consciousness	1.02 (1.01, 1.02) **
Impulsivity	1.03 (1.02, 1.04) ***
Vulnerability	0.98 (0.97, 0.99) ***
Excitement seeking	1.02 (1.01, 1.02) ***
Competence	1.08 (1.07, 1.10) ***
Self-discipline	1.04 (1.03, 1.05) ***
Deliberation	0.96 (0.95, 0.97) ***

** *p*-value < 0.01; *** *p*-value < 0.001; ^1^ adjusted for sociodemographic factors.

**Table 4 ijerph-17-06240-t004:** Associations of personality facets and labor market attachment among individuals with at least one chronic health condition ^1^.

Characteristics	Cancer *N* = 2931	T2DM *N* = 1043	Depression *N* = 8299	Anxiety Disorder *N* = 2658	Heart Disease *N* = 1246	COPD *N* = 4043	Musculoskeletal Disorder *N* = 6026
Age	0.92 (0.91, 0.93) ***	0.92 (0.90, 0.94) ***	0.97 (0.97, 0.98) ***	0.97 (0.96, 0.98) ***	0.90 (0.89, 0.92) ***	0.95 (0.94, 0.96) ***	0.92 (0.91, 0.93) ***
Gender							
Female	Ref	Ref	Ref	Ref	Ref	Ref	Ref
Male	2.60 (2.06, 3.27) ***	2.51 (1.85, 3.40) ***	1.87 (1.66, 2.11) ***	2.25 (1.81, 2.80) ***	2.26 (1.68, 3.03) ***	2.68 (2.25, 3.20) ***	2.61 (2.26, 3.00) ***
Marital status							
Living together	Ref	Ref	Ref	Ref	Ref	Ref	Ref
Not living together	0.96 (0.75, 1.23)	0.66 (0.45, 0.95) *	0.63 (0.56, 0.70) ***	0.70 (0.57, 0.85) **	0.78 (0.55, 1.11)	0.61 (0.50, 0.73) ***	0.69 (0.58, 0.81) **
Educational level							
High	Ref	Ref	Ref	Ref	Ref	Ref	Ref
Medium	0.79 (0.61, 1.01)	0.69 (0.44, 1.09)	0.62 (0.54, 0.71) ***	0.64 (0.50, 0.81) **	0.86 (0.58, 1.27)	0.72 (0.57, 0.91) **	0.72 (0.60, 0.86) ***
Low	0.33 (0.26, 0.42) ***	0.38 (0.25, 0.58) ***	0.34 (0.29, 0.39) ***	0.33 (0.26, 0.43) ***	0.69 (0.48, 0.98) *	0.38 (0.30, 0.48) ***	0.39 (0.33, 0.46) ***
Personality facets							
Anger-hostility	0.96 (0.93, 0.98) **	0.97 (0.93, 1.01)	0.98 (0.97, 0.99) **	0.97 (0.95, 0.99) **	0.98 (0.94, 1.01)	0.97 (0.94, 0.99) **	0.97 (0.95, 0.99) **
Self-consciousness	1.02 (1.00, 1.05)	1.02 (0.98, 1.06)	1.01 (1.00, 1.03) *	1.01 (0.99, 1.04)	1.02 (0.98, 1.06)	1.03 (1.01, 1.05) **	1.02 (1.00, 1.04) *
Impulsivity	1.02 (0.99, 1.05)	1.05 (1.00, 1.10) *	1.03 (1.01, 1.04) ***	1.02 (0.99, 1.04)	1.07 (1.02, 1.12) **	1.05 (1.02, 1.08) ***	1.04 (1.02, 1.06) ***
Vulnerability	0.99 (0.96, 1.02)	0.94 (0.89, 0.99) *	0.98 (0.97, 1.00)	0.97 (0.94, 1.00) *	1.00 (0.96, 1.05)	1.00 (0.97, 1.02)	0.98 (0.95, 1.00) *
Excitement seeking	1.01 (0.98, 1.03)	1.04 (1.01, 1.08) *	1.02 (1.00, 1.03) **	1.02 (0.99, 1.04)	1.03 (1.00, 1.06)	1.02 (1.00, 1.04)	1.02 (1.00, 1.03) *
Competence	1.09 (1.04, 1.13) ***	1.05 (0.99, 1.12)	1.08 (1.06, 1.10) ***	1.07 (1.03, 1.11) ***	1.09 (1.03, 1.16) **	1.12 (1.08, 1.16) ***	1.08 (1.05, 1.11) ***
Self-discipline	1.05 (1.02, 1.08) **	1.02 (0.97, 1.07)	1.04 (1.03, 1.05) ***	1.04 (1.01, 1.06) **	1.06 (1.01, 1.10) **	1.04 (1.01, 1.06) **	1.05 (1.03, 1.07) ***
Deliberation	0.97 (0.94, 1.00) *	0.96 (0.92, 1.01)	0.97 (0.95, 0.98) ***	0.97 (0.95, 1.00) *	0.98 (0.94, 1.02)	0.97 (0.95, 1.00) *	0.96 (0.94, 0.98) ***

^1^ Stratified by chronic disease, and corrected for sociodemographic factors; odds ratios with their corresponding 95% confidence intervals. Abbreviations: T2DM: type 2 diabetes mellitus; COPD: chronic obstructive pulmonary disease. * *p*-value < 0.05; ** *p*-value < 0.01; *** *p*-value < 0.001. Only estimates significant at *p* < 0.01 or smaller were interpreted.

**Table 5 ijerph-17-06240-t005:** Overview of the findings in relation to the a priori expectations.

Personality Factor and Facet Traits		Expected Association with Work Status	Overall	Cancer	T2DM	Depression	Anxiety	Heart Disease	COPD	MSD	Total across CHCs
**Neuroticism**	Prone to negative emotions and pessimism										
Anger-hostility	Angry, irritable, frustrated, and bitter	−	−	−		−	−		−	−	5
Self-consciousness	Prone to shame, embarrassment, and feeling uncomfortable and inferior in company	−	+						+		1
Impulsivity	Inability to control cravings and urges	−	+			+		+	+	+	4
Vulnerability	Dependent, hopeless and panicky when stressed	−	−								0
**Extraversion**	Sociability, energy and optimism										
Excitement seeking	Search for new experiences and feelings	+	+			+					1
**Conscientiousness**	Organized, planful, and self-disciplined										
Competence	Capable, sensible, prudent, and effective	+	+	+		+	+	+	+	+	6
Self-discipline	Task focus, perseverance, and self-control	+	+			+	+	+	+	+	5
Deliberation	Cautious, deliberate, and considerate	+	−			−				-	2
**Total within specific CHCs**			8	2	0	6	3	3	5	5	

Note: Descriptions derived from Costa and McCrae (2006). Low scores can be inferred as the reverse/opposite of a high score. Data on the facets included in this table were the only available personality traits in the Lifelines Cohort Study; see method section for further details. The ‘–’ indicates that the higher the scores on these scales the more likely these people are to become unemployed when faced with a chronic health condition (CHC), whereas ‘+’ means that higher scores are expected to be associated with a higher odds of being employed while having a CHC. Abbreviations: T2DM: type 2 diabetes mellitus; COPD: chronic obstructive pulmonary disease; MSD: musculoskeletal disorder; CHC: chronic health condition; +: positive association; −: negative association.

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
