# Peer review of "Personality as a Resource for Labor Market Participation among Individuals with Chronic Health Conditions"

_ijerph, 2020, doi:10.3390/ijerph17176240_

Round 1

Reviewer 1 Report

As a reviewer for this journal, I have read many manuscripts and this one is one of the most outstanding papers.
First, the topic of this paper is really interesting. This paper seeks to explore the relationships between personality traits and employment status in people with chronic health conditions. The research gap is clearly presented. In this case, the importance of this paper is shown to the reader.
Second, the methods are clearly described. Relevant data are presented.
Third, the results are well explained. The authors also discuss the inconsistency of hypothesized directions and actual findings. I would suggest the authors give some future research directions for scholars.
Out of curiosity, are there any studies using this dataset suggesting a link between certain chronic health conditions and personality traits?

Reviewer 2 Report

The article is interesting and contains interesting content in scientific psychological and medical.It contains valuable content for labour  psychology.

1)Title - is it not worth shortening it? define a shorter term?
2)The authors examine personality traits and not the whole personality structure with one test? It is worth presenting a precise psychological definition of personality variables - the authors examine psychological variables - personality traits. It is as a variable consisting of many features, factors? In my opinion, the empirical definition should be given precisely because personality has many definitions in psychological theories in various psychological paradigms (psychoanalytic, behavioural-cognitive, humanistic, existential, developmental, etc. ). 3)A very broad spectrum of life age of the examined persons (18-65 years) - it is worth specifying, explaining the meaning - why such a period of life was chosen (the personality of an 18-year-old person differs from the personality and resources of a 65-year-old person due to the age of life and other life experiences, autobiographical? 4)Explain why the survey was conducted in 2013 and is currently being published? why was no further research data collected in later years? - it's worth explaining.  5) Measurement of variables - chronic diseases - only on a nominal scale (yes / no) - I understand that the variable was checked for a verified general medical diagnosis in the respondents?  6) Conclusions are general - it is worth to indicate how to apply these results in clinical psychological, psychiatric, psychotherapeutic practice in a specific way.

Reviewer 3 Report

This is an interesting topic; however, I have some concerns and provide some comments that might help the authors to develop their paper.

  1. Introduction

The authors focus on how personality traits relate to organizational variables such as acceptance and problem-solving. Then, they base the study that there is no literature on the link between facet-level personality traits and employment status in people with CHC. There is certainly little literature on this subject, but there is. I feel that the existing research in the field and the research gap are not sufficiently developed. In its current form the introduction seems imprecise and lacks depth and details in the arguments you build. It would be important to argue this issue deeper.

https://academic.oup.com/psychsocgerontology/article/68/6/912/656920

https://www.researchgate.net/profile/Sara_Weston/publication/276853588_Personality_Traits_Predict_the_Onset_of_Disease/links/5647a99d08ae451880ac505c.pdf

https://www.researchgate.net/profile/Sabrina_Paterniti/publication/11232061_Psychosocial_factors_at_work_personality_traits_and_depressive_symptoms_Longitudinal_results_from_the_GAZEL_Study/links/5407771a0cf2c48563b47d5b.pdf

2.1. Design and sample

The authors analyze as covariates age, gender, marital status, and educational level. It would be important that in paragraph 2.1. authors specify not only the age range of the entire sample, but also the mean and standard deviation. Authors should also specify ratio of a) men and women, b) persons living together (marital status) and c)  persons (or ratio) in each level of education. It would also be interesting if they provided some more data as a professional sector, occupational groups, job status,   

Discussion

La discusión está muy bien estructurada. Results relative to the variables Self-consciousness, impulsivity and deliberation can be argued better. For example, they might be moderated by the type of profession, the job status
